# Energy Storage Properties of Sol–Gel-Processed SrTiO_3_ Films

**DOI:** 10.3390/ma16010031

**Published:** 2022-12-21

**Authors:** Jinpeng Liu, Ying Wang, Xiao Zhai, Yinxiu Xue, Lanxia Hao, Hanfei Zhu, Chao Liu, Hongbo Cheng, Jun Ouyang

**Affiliations:** 1Institute of Advanced Energy Materials and Chemistry, School of Chemistry and Chemical Engineering, Qilu University of Technology (Shandong Academy of Sciences), Jinan 250353, China; 2School of Physics, Shandong University, Jinan 250100, China; 3Key Laboratory for Liquid-Solid Structure Evolution and Processing of Materials (Ministry of Education), School of Materials Science and Engineering, Shandong University, Jinan 250061, China; 4Key Laboratory of Key Film Materials & Application for Equipments (Hunan Province), School of Material Sciences and Engineering, Xiangtan University, Xiangtan 411105, China; 5Hunan Provincial Key Laboratory of Thin Film Materials and Devices, School of Material Sciences and Engineering, Xiangtan University, Xiangtan 411105, China

**Keywords:** energy storage, SrTiO_3_, thin films, annealing temperature, sol–gel

## Abstract

Dielectric films with a high energy storage density and a large breakdown strength are promising material candidates for pulsed power electrical and electronic applications. Perovskite-type dielectric SrTiO_3_ (STO) has demonstrated interesting properties desirable for capacitive energy storage, including a high dielectric constant, a wide bandgap and a size-induced paraelectric-to-ferroelectric transition. To pave a way toward large-scale production, STO film capacitors were deposited on Pt(111)/Ti/SiO_2_/Si(100) substrates by the sol–gel method in this paper, and their electrical properties including the energy storage performance were studied as a function of the annealing temperature in the postgrowth rapid thermal annealing (RTA) process. The appearance of a ferroelectric phase at a high annealing temperature of 750 °C was revealed by X-ray diffraction and electrical characterizations (ferroelectric *P*-*E* loop). However, this high dielectric constant phase came at the cost of a low breakdown strength and a large hysteresis loss, which are not desirable for the energy storage application. On the other hand, when the RTA process was performed at a low temperature of 550 °C, a poorly crystallized perovskite phase together with a substantial amount of impurity phases appeared, resulting in a low breakdown strength as well as a very low dielectric constant. It is revealed that the best energy storage performance, which corresponds to a large breakdown strength and a medium dielectric constant, is achieved in STO films annealed at 650 °C, which showed a large energy density of 55 J/cm^3^ and an outstanding energy efficiency of 94.7% (@ 6.5 MV/cm). These findings lay out the foundation for processing high-quality STO film capacitors via the manufacturing-friendly sol–gel method.

## 1. Introduction

As environmental-friendly energy storage devices, dielectric capacitors have attracted extensive interests due to their fast charging and discharging speed and high power density [1,2]. With the rapid development of electrical and hybrid vehicles, mobile electronics and high-energy laser weapons [3,4,5,6], dielectric capacitors have become increasingly important as key components in these pulsed power systems. Compared with conventional ceramic capacitors, thin film ferroelectric capacitors exhibit a higher energy storage density and a larger breakdown electric field. The energy storage density and efficiency (*η*) for a ferroelectric capacitor can be calculated from its polarization–electric field (*P*-*E*) loop using the following equations [3,6,7,8]:(1)$$W_{rec}=\int_{P_{r}}^{P_{max}}EdP$$
(2)$$W_{c}=\int_{0}^{P_{max}}EdP$$
(3)$$\eta =\frac{W_{rec}}{W_{c}}$$

In these equations, *W_rec_* and *W_c_* are the recyclable energy density and charged energy density, respectively, *E* is the external electric field, and *P_r_* and *P_max_* are the remnant and maximum electric polarizations, respectively.

The rich properties of SrTiO_3_ (STO), including a wide bandgap, a high dielectric constant, diamagnetic and thermal stabilities, etc., have created a broad spectrum of applications in a variety of research fields, including high-k gate layers, dynamic random access memories, microwave components, varistors, etc. [9,10,11,12,13,14]. Specifically, as a typical quantum paraelectric, STO not only possesses a high dielectric constant of ~300, but also shows a low dielectric loss and an excellent thermal stability [14]. These features demonstrate a great promise for STO to be used in energy storage dielectric capacitors. At present, there have been some studies on energy storage characteristics of STO-based thin films with a complex composition or structure, created by the methods of multielement doping or solid solutioning [15,16,17,18,19,20,21]. For example, Diao et al. reported a large recoverable energy storage density of 23.8 J/cm^3^ with an efficiency of 69.8% in Mn-modified SrTiO_3_ thin films [22]. Khassaf et. al. studied the effect of interlayer coupling on the polarization response of a PbZr_0.2_Ti_0.8_O_3_/SrTiO_3_ bilayer and reported an excellent energy storage capability achieved in this system [23]. Moreover, Pan et al. revealed that enhanced energy storage properties could be achieved in 0.4BiFeO_3_-0.6SrTiO_3_ solid-solution films deposited on Nb-doped STO (Nb:STO) single crystalline substrates, including a large recoverable energy density of 70 J/cm^3^, by suppressing the formation of oxygen vacancies [24].

Apparently, although these above methods can improve energy storage performances of the STO thin films, the complex composition or film structure, as well as the costly single crystalline substrates being used, have inevitably increased the difficulties for practical applications of these film materials [25,26,27,28,29,30]. In addition, as far as we know, there are very few reports on the energy storage properties of pure STO thin films [14]. In this work, pure STO thin films were deposited on Pt(111)/Ti/SiO_2_/Si(100) substrates by a simple sol–gel method. We focus on the effects of annealing temperature on the microstructure, leakage current, dielectric and ferroelectric properties of the STO films. By selecting an appropriate annealing temperature, we achieved a pseudolinear *P*-*E* loop with a large maximum polarization and a small remanent one. This can be attributed to a well-controlled nanograin size. This finding has significantly enhanced the energy storage performance of sol–gel-processed STO films.

## 2. Experimental Section

In this work, STO thin films were prepared by using a sol–gel method. Strontium acetate hemihydrate (Sr(CHOO)_2_∙0.5H_2_O)) and tetrabutyl titanate (Ti[OCH(CH_3_)_2_]_4_) were chosen as raw materials, and 2-methoxyethanol, acetylacetone and acetic acid were used as solvents to prepare the precursor with a concentration of 0.25 M. Specifically, the strontium acetate hemihydrate and the tetrabutyl titanate were dissolved in acetic acid and 2-methoxyethanol, respectively. Strontium acetate was initially dissolved into hot acetic acid (60 ℃) in a beaker under constant stirring for half an hour and sealed with plastic wrap, leading to the formation of a transparent and colorless solution. Meanwhile, titanium tetra-butoxide was mixed with 2-methoxyethanol in another beaker at room temperature under constant stirring. Note that a small amount of acetylacetone was added to stabilize the tetrabutyl titanate solution. The above two solutions were mixed under constant stirring for 6 h and then aged for 24 h to obtain the final precursor solution. The film depositions were carried out by spin coating at 4500 rpm for 15 s on Pt(111)/Ti/SiO_2_/Si substrates. After each deposition, drying and pyrolysis of the film were performed on a hot plate at 150 °C for 3 min and 450 °C for 5 min, respectively. The spin coating deposition–drying–pyrolysis process cycle was repeated 8 times to obtain an amorphous STO film with a desired thickness (~300 nm). Lastly, the amorphous STO thin films were annealed in a pure oxygen atmosphere via a rapid thermal annealing (RTA) process at 550 °C, 650 °C and 750 °C for 3 min, respectively. It is noted that RTA is an extremely fast heating method that can reach a high temperature in tens of seconds. Unlike the conventional furnace annealing process, RTA usually takes only 3~5 min to crystallize the amorphous films. This has been reported in a large number of reports in the literature [31,32,33,34,35]. The detailed process flow chart is shown in Figure 1 below.

The films’ phase structures and crystallographic orientations were examined using XRD *θ*-2*θ* scans (Rigaku Dmax-2500PC XRD diffractometer (Tokyo, Japan) equipped with a Ni-filtered Cu Ka radiation source, Smartlab SE. The surface and cross-sectional film morphologies, as well as elemental compositions, were characterized using a field-emission scanning electron microscope (ZEISS Gemini 300, Oberkochen, Germany) equipped with energy-dispersive X-ray spectroscopy (EDX). The surface morphologies of the films were recorded using an Atomic Force Microscope (AFM, MFP-3D Infinity, Asylum Research, Santa Barbara, CA, USA) with a Ti/Ir-coated conductive ASYELEC.01-R2 probe. Prior to the electrical measurements, circular Au top electrodes (200 μm in diameter) were deposited on the STO films via a shadow mask. The electric polarization hysteresis loops (*P*-*E* loops) and leakage current characteristics of the STO films were measured using a Precision Premier II ferroelectric tester (Radiant Technologies). The frequency-dependent dielectric properties were examined using a precision LCR meter (TH2838H, Tonghui, Changzhou, China).

## 3. Results and Discussion

Figure 2 shows the XRD patterns of the STO films annealed at different temperatures, and a randomly oriented polycrystalline structure was revealed. Clearly, the STO film annealed at a low temperature of 550 °C shows a poor crystallinity with some secondary phases, which include nonstoichiometric strontium oxide (Sr_3_Ti_2_O_7_, Sr_2_Ti_6_O_13_) and titanium oxide (TiO_x_). With an increasing annealing temperature, an enhanced crystallization of the film is observed. The STO film annealed at 750 °C shows the strongest and sharpest diffraction peak, as well as the lowest number of impure phases. The grain size (*D*) of the STO films was analyzed by using the Scherrer equation below [36]: (4)$$D=\frac{K\lambda }{\beta \;cos\theta }$$
where *K* is a constant (*K* = 0.89), *λ* is the wavelength of the X-ray, *β* is the diffraction peak’s full width at half maximum height (FWHM) and *θ* is the diffraction angle. It is noted that the STO (200) peaks are used for this analysis. Consistent with the qualitative analysis above, *D* grows with the annealing temperature, which is computed to be ~10 nm, ~20 nm and ~40 nm for the STO films annealed at 550 °C, 650 °C and 750 °C, respectively. The different grain sizes and degrees of crystallinity of the STO films have significant impacts on their electrical properties, especially the energy storage characteristics. These impacts will be discussed in detail in subsequent sections.

Figure 3a shows the SEM surface morphology of the STO films deposited on Pt-coated Si and annealed at 650 °C. Clearly, a flat surface with densely packed nanograins is observed and there are no visible cracks. The cross-sectional SEM image of the film/electrode/substrate heterostructure, shown as the inset of Figure 3a, also allows a direct measurement of the layer thicknesses. The STO and Pt layers were measured to be ~300 nm and ~350 nm thick, respectively. Energy dispersive X-ray spectroscopy (EDS) studies were performed to investigate the chemical composition of the STO film. The EDS mappings of each element displayed in Figure 3b–d confirm the homogeneous distributions of the Sr, Ti and O elements. Moreover, as clearly shown in Figure 4a, only Sr, Pt, Si, Ti and O elements were detected from the STO/Pt/Ti/SiO_2_/Si thin film heterostructure, and a 1:0.8 chemical stoichiometric ratio of Sr/Ti was revealed. This small deviation from the chemical stoichiometry can be attributed to the semiquantitative nature of the EDS analysis, as well as some diffusion/evaporation losses of impurity species containing Ti [7,19,22,25]. It should be noted that the other two STO films have shown similar surface morphologies and chemical composition. To further understand the effect of annealing temperature on the surface morphologies of STO films, we performed atomic force microscopy (AFM) analyses of the STO films. As shown in Figure 4b, the film annealed at 650 °C showed a fine-grained surface morphology with a root mean square (RMS) surface roughness of ~4.9 nm. Meanwhile, the film annealed at 750 °C (Figure 4c) displayed larger grains and a smoother surface, with an RMS surface roughness of ~0.6 nm. These observations are consistent with the XRD results.

Figure 5a gives the leakage current density (*J*-*E*) curves of the STO films measured in the electric field range of –800 to 800 kV/cm. Compared with that of the film annealed at 750 °C at 800 kV/cm (1.1 × 10^−2^ A/cm^2^), leakage current densities reduced by nearly four orders of magnitude at 800 kV/cm were observed in the STO films annealed at 650 °C (4.5 × 10^−6^ A/cm^2^) and 550 ℃ (1.2 × 10^−6^ A/cm^2^), respectively, indicating high electrical resistances for the latter two. To further understand the above leakage current characteristics, dielectric constants and losses of the three STO films as a function of frequency ranging from 10 kHz to 2 MHz were measured, and the results are shown in Figure 5b. Compared with the films annealed at 650 °C and 550 °C, a higher dielectric constant was achieved in the STO film annealed at 750 °C, which can be attributed to its higher degree of crystallinity and larger grain size [2]. Moreover, consistent with the leakage current characteristics, the STO films annealed at 650 °C and 550 °C display lower losses than that of the STO film annealed at 750 °C, when the measuring frequency exceeds 50 kHz. The higher dielectric losses near the low-frequency end for the 550 °C annealed film can be attributed to its poor crystallinity and substantial number of impurity phases. These crystalline defects are most likely accumulated near the interface of the film, leading to a prominent dielectric relaxation in the low-frequency range [29]. On the other hand, the STO film annealed at 650 °C shows a reduced loss throughout the measuring frequency range, indicating that interfacial and bulk defects were both significantly reduced in this film. These observations are consistent with the XRD results. Specifically, the dielectric constants and losses were measured at 100 kHz are 95 and 0.12, 14 and 0.023 and 6 and 0.052 for the STO films annealed at 750 °C, 650 °C and 550 °C, respectively. It is worth noting here that the dielectric characteristics of the STO film annealed at 650 °C shows almost frequency-independent behavior over a broad frequency range, i.e., its dielectric constant and loss tangent exhibits a good frequency stability.

Room temperature unipolar *P*-*E* loops (@ 1 kHz) of the STO films annealed at different temperatures are presented in Figure 6. The STO film annealed at 750 °C shows clear ferroelectric characteristics, which include a large remnant polarization and a sizable hysteresis loop. These features are better illustrated by its bipolar *P*-*E* loop (shown in the inset). The ferroelectric property observed in the 750-annealed STO thin film is closely related to the compressive strain and its increased grain size from the XRD pattens [37,38]. On the other hand, slim pseudolinear *P*-*E* loops with a small remnant polarization (*P_r_*) and a reduced hysteresis loop are achieved in STO films annealed at 650 °C and 550 °C. In particular, the STO film annealed at 650 °C exhibits an elongated, tilted *P*-*E* loop with a *P_max_* of ~18 μC/cm^2^ and a *P_r_* of ~0.5 μC/cm^2^, both being achieved under a large maximum applicable electric field of *E_max_* = 6.5 MV/cm. These characteristics endow the film with an excellent energy storage capability. In the charging process of a dielectric capacitor, the polarization/electric displacement *P* increases from zero to its maximum *P_max_* as the applied electric field *E* increases from zero to *E_max_*. The capacitive energy stored per unit volume can be calculated by $W_{c}=\int_{0}^{P_{max}}EdP\;$[3]. On the other hand, during the discharging process, the polarization/electric displacement reduces from *P_max_* to *P_r_* as the electric field across the capacitor decreases from *E_max_* to zero. This part of stored energy that is released upon discharge, i.e., the recoverable/recyclable electric energy (density) *W_rec_*, is calculated by $W_{rec}=\int_{P_{r}}^{P_{max}}EdP\;$[3]. This energy term is represented by the shaded area in Figure 6 for the 650 °C-processed STO films. Lastly, the energy storage efficiency, i.e., charge–discharge efficiency *η*, is defined by $\eta =\left(W_{rec}/W_{c}\right)\times 100\%\;$[3]. Based on the above definitions, the recoverable energy density (*W_rec_*) and the efficiency (*η*) of the STO film annealed at 650 °C are calculated to be 55 J/cm^3^ and ~94.7%, respectively. These results are significantly improved against previously reported ones, as they are compared head-to-head in Figure 6 below.

Figure 7 summarizes the recoverable energy density (*W_rec_*), efficiency (*η*) and breakdown field strength (*E*_b_) of our STO film annealed at 650 °C, in direct comparisons with some representative STO-based films including doped, bilayer or multilayer films deposited on Si substrates [7,16,17,20,22,26,38,39,40]. Clearly, our results are superior to those STO-based films previously studied. With such large *W_rec_* and *η* values, as well as an extremely high *E*_b_ and a simple composition, our sol–gel-processed STO films show a great potential in high-performance dielectric capacitors.

## 4. Conclusions

In this work, STO thin films were deposited on Pt/Ti/SiO_2_/Si substrates via a sol–gel process. Effects of annealing temperature in a postgrowth RTA process on the microstructures and energy storage characteristics of the STO films were discussed. By adjusting the annealing temperature to 650 °C, a pseudolinear *P*-*E* loop with a high energy storage performance was achieved in the STO film. Enhanced electrical properties, including a medium dielectric constant and a low dielectric loss, a low leakage current, a high breakdown strength, as well as a large maximum polarization and a small remanent one, were obtained in the 650 °C-annealed STO film. These features endow the film with the best energy storage characteristics, with a large recyclable energy density of 55 J/cm^3^ and an outstanding energy efficiency of 94.7% (@ 6.5 MV/cm). These findings lay out a foundation for the application of STO film capacitors in high-performance dielectric energy storage.

## Figures and Tables

**Figure 1 materials-16-00031-f001:**
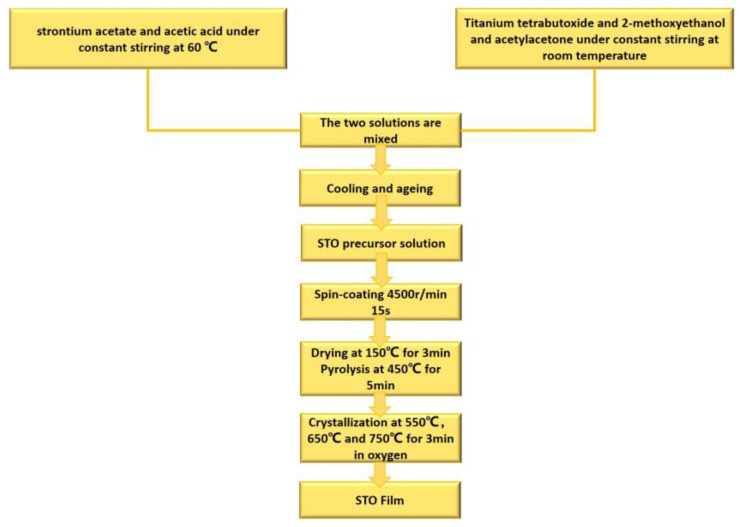
Flow chart for the preparation of the STO films in a sol–gel process.

**Figure 2 materials-16-00031-f002:**
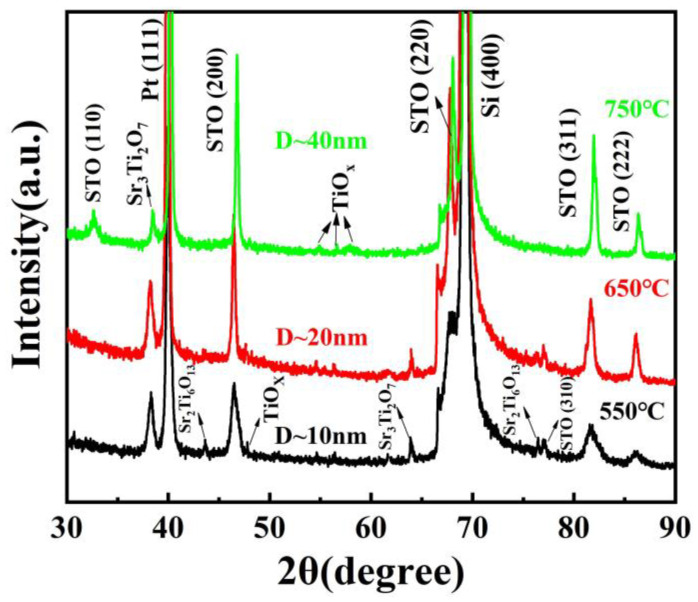
XRD patterns of the STO thin films annealed at different temperatures.

**Figure 3 materials-16-00031-f003:**
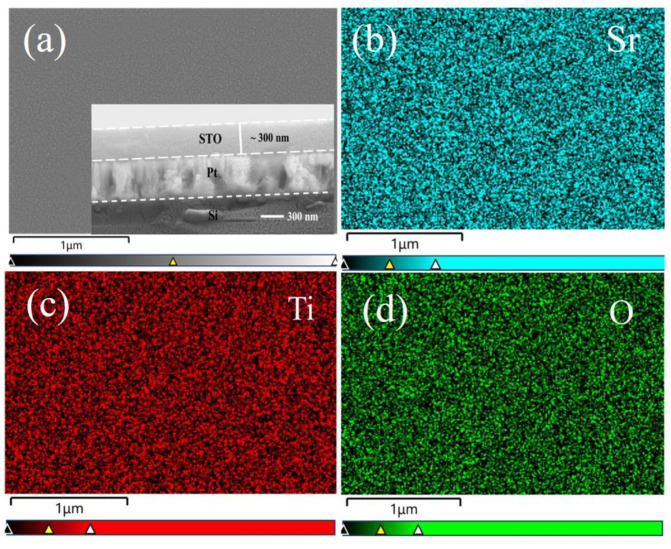
SEM characterizations of the STO thin film at 650 °C: (**a**) surface and cross-sectional morphologies, (**b**–**d**) EDS mappings of Sr, Ti and O elements.

**Figure 4 materials-16-00031-f004:**
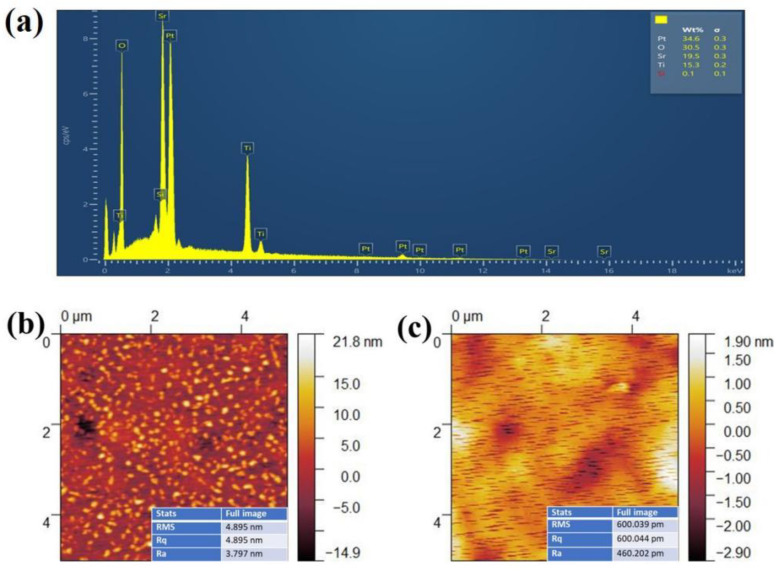
(**a**) Quantitative elemental EDS spectra and (**b**,**c**) AFM surface morphologies of the STO thin films annealed at (**b**) 650 °C and (**c**) 750 °C.

**Figure 5 materials-16-00031-f005:**
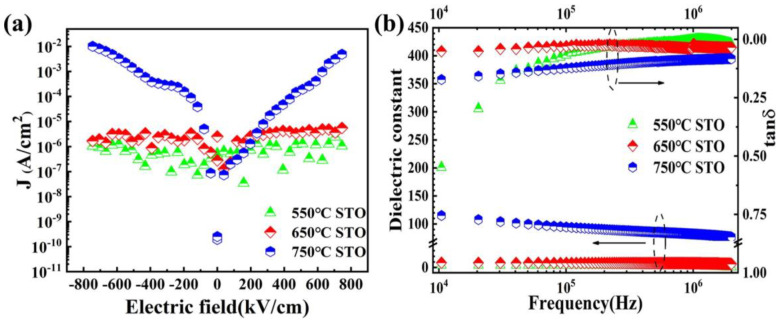
(**a**) Leakage current density vs. electric field (*J*-*E*) curves, (**b**) frequency-dependent dielectric behaviors of STO thin films annealed at different temperatures.

**Figure 6 materials-16-00031-f006:**
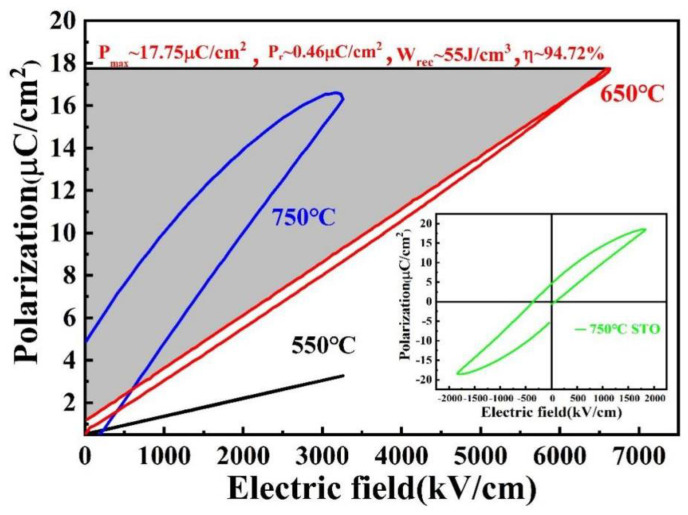
Unipolar *P*-*E* loops of STO films annealed at different temperatures. The inset shows bipolar *P*-*E* loop of the STO film annealed at 750 °C.

**Figure 7 materials-16-00031-f007:**
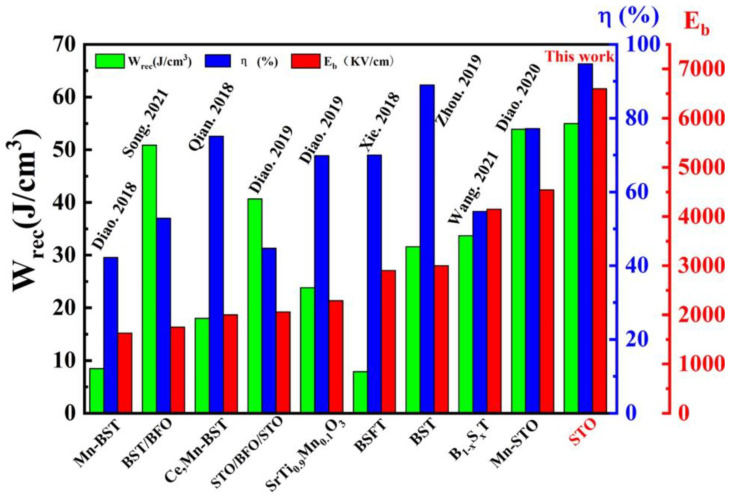
Comparison of energy storage performances between the STO film annealed at 650°C in this work and some previously reported STO-based films deposited on Si substrates [7,16,17,20,22,26,38,39,40].

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
