# Peer review of "Energy Storage Properties of Sol–Gel-Processed SrTiO3 Films"

_materials, 2022, doi:10.3390/ma16010031_

Round 1
Reviewer 1 Report
Specific comments from referee to Author
Dear
Materials
Energy storage properties of sol-gel processed SrTiO3 films
Comments:
The work entitled “Energy storage properties of sol-gel processed SrTiO3 films”. The paper fits the scope of the Materials. However, it is not clear the experimental procedure, is losing some information and is necessary a better discussion the results. In general, needs to improve.
Abstract
1. OK
Introduction
1. It would be interesting to write about SrTiO3 for other applications.
2. Experimental
1. The experimental part is unclear. Maybe making a flow diagram would help a better understanding. This parte must be rewrite.
2. information is missing how the thickness was measured
3. Who means a desired thickness?
4. The justification why just 3 minutes for heat treatment
3 Results and discussions
1. Figure 1 is an a bad resolution
2. Should be interesting the discussion using the results of XRD with SEM, the relationship between this parameter in the dielectric behavior
3. About figure 4 b, is very hard to understand this figure, maybe if you separate in 3 parts.
4. About dielectric constant, is possible to try to fitting this result with some mathematical model
Author Response
Dear editor Octavian Barbos,
First of all, we are grateful to you for sending us the reviewers’ comments and giving us an opportunity to revise our manuscript (materials-2048677), which is entitled “Energy storage properties of sol-gel processed SrTiO3 films”. We have revised the manuscript carefully and made point-to-point responses to these comments, which surely helps us improve the quality of our paper significantly. All the changes in response to these comments have been highlighted in yellow in the revised manuscript for the reviewers’ convenience. Here we present our detailed responses as follows:
Reviewer #1: The work entitled “Energy storage properties of sol-gel processed SrTiO3 films”. The paper fits the scope of the Materials. However, it is not clear the experimental procedure, is losing some information and is necessary a better discussion the results. In general, needs to improve.
Abstract
- OK.
Reply: We appreciate the reviewer for his/her positive evaluation.
Introduction
- It would be interesting to write about SrTiO3 for other applications.
Reply: Thank you very much for the reviewer’s comments. Discussion of some other applications of SrTiO3 have been added in the revised manuscript (lines 51-54).
Experimental
- The experimental part is unclear. Maybe making a flow diagram would help a better understanding. This part must be rewrite.
Reply: We are grateful to the reviewer for his/her constructive comment. Following the suggestion of reviewer, the experimental part has been re-written and an experimental flow chart was added (lines 87-98, Fig. 1).
- information is missing how the thickness was measured.
Reply: We thank the reviewer for his/her valuable comment. The cross-sectional image of the film via SEM was used to determine its thickness, which was given in the revised manuscript (lines 146-149, Fig.3a).
- Who means a desired thickness?
Reply: We thank the reviewer for his/her valuable question and apologize for the confusion. As described in the experimental section (lines 97-98), a desired film thickness is that obtained by repeating the process cycle of spin coating, drying and pyrolysis for 8 times, which is about 300 nm (typical of the films prepared via a sol-gel method).
- The justification why just 3 minutes for heat treatment.
Reply: We thank the reviewer for his/her such valuable question. The justification why just 3 minutes for heat treatment have been discussed in the revised manuscript (lines 101-105).
Results and discussions
- Figure 1 is an a bad resolution
Reply: We thank the reviewer for his/her keen observation and a clearer XRD pattern is shown in Fig. 2 in the revised manuscript.
- Should be interesting the discussion using the results of XRD with SEM, the relationship between this parameter in the dielectric behavior
Reply: We are grateful to the reviewer for his/her constructive comments and fully agree with him/her. The relationship between the results based on XRD and SEM characterizations and the dielectric behaviors of the STO films has been further discussed in the revised manuscript (lines 180-191).
- About figure 4 b, is very hard to understand this figure, maybe if you separate in 3 parts.
Reply: We thank the reviewer for his/her keen observation and apologize for the obscurity Fig. 4b has caused. According to the reviewer's recommendation, the frequency-dependent dielectric constant and loss curves were redrawn in the revised manuscript (Fig.5b).
- About dielectric constant, is possible to try to fitting this result with some mathematical model.
Reply: We thank the reviewer for his/her valuable input. Yes, the reviewer is absolutely right on the possibility of mathematical modeling of the dielectric behavior. However, since dielectric constant is high orientation-dependent, it is possible to do so in an epitaxial or highly oriented STO film, but not in polycrystalline films like in our case.
It should be noted that, according to the suggestion of the reviewer, we have asked a native English speaker to check the language of the manuscript in its entirety, and these changes have been highlighted in yellow in the revised version.
Thank you for your time and consideration. I look forward to hearing from you soon.
Best wishes
Yours sincerely
Hanfei Zhu Jun Ouyang,
Ph.D., Ph.D., Professor,
Institute of Advanced Energy Materials and Chemistry, School of Chemistry and Chemical Engineering, Qilu University of Technology
Daxue Road #3501, Jinan 250353, China
E-mail: zhf@qlu.edu.cn (H. Zhu), E-mail: ouyangjun@qlu.edu.cn (J. Ouyang)

Reviewer 2 Report
Please see the attached file.

Author Response
Dear editor Octavian Barbos,
First of all, we are grateful to you for sending us the reviewers’ comments and giving us an opportunity to revise our manuscript (materials-2048677), which is entitled “Energy storage properties of sol-gel processed SrTiO3 films”. We have revised the manuscript carefully and made point-to-point responses to these comments, which surely helps us improve the quality of our paper significantly. All the changes in response to these comments have been highlighted in yellow in the revised manuscript for the reviewers’ convenience. Here we present our detailed responses as follows:
Reviewer #2:
The author investigated energy storage device based on STO films with large energy density and very high energy efficiency. Details material characterization has been demonstrated. All pictures are well organized. While few doubts are puzzling me.
- There is no doubt that cubic bulk SrTiO3 are paraelectric, not ferroelectric.
Only in some extreme conditions such highly strained, SrTiO3 thin films can show ferroelectric property. The author stated that >300nm SrTiO3 thin films show ferroelectric PE loops after 750C annealing, what is the origin of ferroelectricity here?
Reply: We thank the reviewer for his/her such valuable comment and the ferroelectric property of the 750 °C-annealed STO thin film was brief discussed in the revised manuscript (lines 204-207). We agree with his/her point that the cubic bulk SrTiO3 is paraelectric, not ferroelectric. In fact, the STO thin film annealed at 750 °C in this study showed a compressive residual strain of about -0.6% via XRD analysis. As the reviewer commented, this effect has an important impact on the paraelectric-ferroelectric phase transition of the film. In fact, the ferroelectric behavior shown by the compressively strained STO film (annealed at 750 °C) is completely consistent with the research works reported by Haeni and Ren et al. [R1, R2]. On the other hand, as shown by the XRD pattern in Fig. 2 in this study, the large grains of the STO film were achieved when annealed at a high temperature crystallization of 750 °C. When a sufficient electric field was applied (3 MV/cm in present case), the polarization switching was thus induced. These result are well consistent with some reported works [R3,R4].
- What is the science behind the experimental results that SrTiO3 can have such high energy density and high energy efficiency? There is only experimental demonstration but barely the discussion of physics behind phenomenon. The only reason I notice is the grain size difference between differently annealed films, but this is not explaining why density and efficiency in this work is better than others examples in Figure 6.
Reply: We are grateful to the reviewer for his/her such constructive comments. The high energy density and efficiency achieved in the STO film annealed at 650 °C is due to its high breakdown strength and slim pseudo-linear P-E loop with a large maximum polarization, as the reviewer pointed out, which is closely related to a synergistic effect with its moderate grain size, crystallization degree and compression strain. In fact, in this study, the effects of the crystallization degree, grain size, impurity phase, dielectric loss and leakage current on the ferroelectric and energy storage characteristics of the STO thin films are mainly discussed, and these detailed discussions can be found in the revised manuscript (lines 168-177). We also agree with the reviewers' comment on the lack of the deeper physics behind this phenomenon. Therefore, in the future study, we will explore the physical mechanism of excellent energy storage properties of the STO thin films based on their detailed microstructures.
- Figure 3b and Figure 3c showed the surface of STO. what are the particles in figure 3b and the stripe-like wells in figure 3c?
Reply: We thank the reviewer for his/her keen observation and put forward the valuable question. As revealed by the planar and 3D AFM images of 650 °C-annealed STO film shown in Fig. R2a and b, the white particles in Fig. 4b in the revised manuscript represent the fully-grown and elongated columnar nanograins of the film. While from the planar and 3D AFM images of the 750 °C-annealed STO film shown in Fig. R2c and d, it is found that these stripe-like wells are not observed in the 3D image, indicating that the stripe-like wells are most likely noises during the AFM probe scanning.
Fig. R2. (a), (c) Planar and (b),(d) 3D AFM surface morphology images of 650 °C-annealed and 750 °C-annealed STO films, respectively.
References
[R1] Haeni, J. H.; Irvin, P.; Chang, W.; Uecker, R.; Reiche, P.; Li, Y.L.; Choudhury, S.; et al. Room-temperature ferroelectricity in strained SrTiO3. nature. 2004, 430, 758-761.
[R2] Le, T.; Kurt, O.; Ouyang, J.; Wang, J.; Chen, L-Q.; Lin, EL.; Ekerdt, JG.; Ren, Y. Engineering nanoscale polarization at the SrTiO3/Ge interface. Scripta Materialia. 2020, 178, 489-492.
[R3] Lee, D.; Lu, H.; Gu, Y.; Choi, SY.; Li, SD.; Ryu, S.; Paudel, TR.; Song, K.; Mikheev, E.; Lee, S.; et al. Emergence of room-temperature ferroelectricity at reduced dimensions. Science. 2015, 349, 1314-1317.
[R4] Hou, C.; Huang, W.; Zhao, W.; Zhang, D.; Yin, Y.; Li, X. Ultrahigh Energy Density in SrTiO3 Film Capacitors. ACS Appl Mater Interfaces. 2017, 9, 20484-20490.
Thank you for your time and consideration. I look forward to hearing from you soon.
Best wishes
Yours sincerely
Hanfei Zhu Jun Ouyang,
Ph.D., Ph.D., Professor,
Institute of Advanced Energy Materials and Chemistry, School of Chemistry and Chemical Engineering, Qilu University of Technology
Daxue Road #3501, Jinan 250353, China
E-mail: zhf@qlu.edu.cn (H. Zhu), E-mail: ouyangjun@qlu.edu.cn (J. Ouyang)
